# Deep Reinforcement Learning With Adaptive Combined Critics

## Abstract

The overestimation problem has long been popular in deep value learning, because function approximation errors may lead to amplified value estimates and suboptimal policies. There have been several methods to deal with the overestimation problem, however, further problems may be induced, for example, the underestimation bias and instability. In this paper, we focus on the overestimation issues on continuous control through deep reinforcement learning, and propose a novel algorithm that can minimize the overestimation, avoid the underestimation bias and retain the policy improvement during the whole training process. Specifically, we add a weight factor to adjust the influence of two independent critics, and use the combined value of weighted critics to update the policy. Then the updated policy is involved in the update of the weight factor, in which we propose a novel method to provide theoretical and experimental guarantee for future policy improvement. We evaluate our method on a set of classical control tasks, and the results show that the proposed algorithms are more computationally efficient and stable than several existing algorithms for continuous control.

## Introduction

The task of deep reinforcement learning (DRL) is to learn good policies by optimizing a discounted cumulative reward through function approximation. Although a variety of control tasks have gained success through DRL Mnih et al. (2013; 2015); Van Hasselt et al. (2016); Wang et al. (2015); Schaul et al. (2015); Lillicrap et al. (2015); Mnih et al. (2016), there still exists biases caused by function approximation errors, studied in prior work Mannor et al. (2007).

Overestimation bias is rooted in Q-learning by consistently maximizing a noisy value estimate, and was originally reported to be present in the algorithms, typically, like deep Q-network (DQN) Mnih et al. (2015) which is aimed for discrete control tasks. Since DQN adopts neural networks to approximate a cumulative future reward, it is unavoidable that the noise evolved from function approximation accompanies the whole training. Besides, DQN updates its policy by choosing the action that maximizes the value function, which may bring the inaccurate value approximation that outweighes the true value, i.e., the overestimation bias. This bias further accumulates at every step via bootstrapping of temporal difference learning, which estimates the value function using the value estimate of a subsequent state.

When the overestimation bias is harmful to a certain task, it will cause instability, divergence and suboptimal policy updates. To solve this, Double deep Q-networks (DDQN) Van Hasselt et al. (2016) alter the policy update strategy by taking actions based on a relatively independent target value function instead of maximizing the original Q network. DDQN not only yields more accurate value estimates, but leads to much higher scores on several games Van Hasselt et al. (2016). However, due to the randomness of approximation errors, DDQN provides no theoretical bounds for the overestimation, and the unresolved overestimation or induced underestimation makes the performance of DDQN worse than DQN in some cases, reported in Van Hasselt et al. (2016).

For the case of high-dimensional, continuous action spaces, Deep Deterministic Policy Gradient (D-DPG) Lillicrap et al. (2015) provide a model-free, off-policy actor-critic method using deterministic policy gradients. Due to the slow-changing policy in an actor-critic setting, the current and target value estimates will be too dependent to circumvent bias if the solution of DDQN to overestimation is directly under the continuous control setting. Inspired by Double Q-learning Hasselt (2010)

which employ a pair of independently trained critics to achieve a less biased value estimation, the twin delayed deep deterministic policy gradient algorithm (TD3) Fujimoto et al. (2018) propose a clipped Double Q-learning variant choosing the lower target value as an approximate upper-bound to estimate the true current value, which favors underestimation biases that will not accumulate during training because actions with high value estimation are preferred. However, there are two problems lying in TD3. First, by taking actions towards the maximization of lower action-value (Q-value) at every step, the policy improvement cannot be guaranteed, which may cause potential suboptimal policies and instability. Second, using the same target value to update two critic networks will make them less independent.

The paper has the following contributions. First, we propose a combined value of two independent critics connected by a weight factor, and use it to update the policy instead of serving as a shared target estimate for the two critics, to avoid losing independence. Second, we propose a sign multiplier which determines whether the updated combined value has increased. The objective function for updating the weight factor is the product of the sign and the combined value evaluated by the updated policy. Third, we present a novel algorithm for continuous control, which can minimize the overestimation bias while providing guarantee for future policy improvement. And theoretical proofs show that the proposed algorithm has the property of asymptotical convergence and expected policy improvement. Fourth, we further apply the proposed algorithm to an unbiased framework to create another algorithm, which can remove the systematic bias due to the probability mismatch between the behavior policy and the target policy existing in off-policy methods. Fifth, extensive evaluations are conducted to compare the proposed algorithms with some baseline algorithms, in terms of computational efficiency, convergence rate and stability.

## BACKGROUND

The task of reinforcement learning (RL) is to learn an optimal policy that maximizes a single return, i.e., the expected discounted cumulative reward of an episode. During an episode, the agent continually receives a sequence of observations by interacting with the environment before encountering an terminated state or the timeout. The return is calculated as the expected sum of future rewards. DRL combines the neural networks with RL so that the return can be approximated by the parameterized function. In DRL, the agent follows a behavior policy to determine future rewards and states, and store these observations in a memory, which will be randomly sampled over to train the network parameters and update the target Q-values. The target updates can be immediate, delayed or "soft".

Generally, the Q-value of DRL is represented as the expected discounted cumulative reward, an estimate function with respect to the state and action, which is given by

$$Q_\pi(s, a) = \mathbb{E}_{p^\pi(h|s_0, a_0)} \left[ \sum_{t=0}^{\infty} \gamma^t r(s_t, a_t) | s_0 = s, a_0 = a \right],$$ (1)

where $r(s, a)$ is the immediate reward which is usually connected to the state-action pair, $(s, a)$ is the value of initial state-action pair, and $\gamma \in (0, 1)$ is the discount horizon factor for future rewards. Under the guidance of behavior policy $\pi$, $p^\pi(h|s_0, a_0)$ is the joint probability of all state-action pairs during an episode given the initial state-action pair $(s_0, a_0)$.

When neural networks are used to approximate Q-values, the update of behavior policy $\pi$ is closely related to the Q network under the setting of Markov Decision Process (MDP). The Q-value function in (1) takes the state-action pair $(s, a)$ as input and maps it to the Q-value. The foundation for updates of network parameters is the Bellman equation. Most of existing DRL algorithms adopt an independent target Q network to approximate an essential part of the target value, which can be set based on Bellman equation and organized to form the general loss function for DRL as Lillicrap et al. (2015)

$$L(\omega) = \mathbb{E}_{(s,a,r,s')} \left[ (r + \gamma Q(s', \mu(s'|\theta')|\omega') - Q(s, a|\omega))^2 \right],$$ (2)

where $a$ is the action drawn from a behavior policy based on $s$, $r$ and $s'$ are the immediate reward and next state received by interacting with the MDP environment, respectively. Overall, $(s, a, r, s')$ is the tuple stored in the replay buffer at every step. Besides, $\mu(s'|\theta')$ is the target policy mapping $s'$ to the next action $a'$ through a target actor network parameterized by $\theta'$ in deterministic actor-critic

methods, instead of taking actions from the replay buffer. Moreover, $\omega$ is the parameter of current Q network, which is normally different from the target network parameter $\omega'$.

Once the Q networks are updated, the objective at the current iteration is to optimize the actor parameter $\theta$, which is updated by maximizing the expected return $J(\theta) = \mathbb{E}_s [Q_\pi(s, \mu(s|\theta)|\omega)]$. In the case of continuous control, $\theta$ can be updated by gradient descent $\nabla_\theta J(\theta)$.

## ADAPTIVE DELAYED DEEP DETERMINISTIC POLICY GRADIENT

In TD3, the clipped variant of Double Q-Learning is proposed for actor-critic to reduce the overestimation bias. Instead of using a pair of actors and critics to learn twice, TD3 upper-bounds the less biased value estimate by taking the minimum between two critic networks to offer target estimation for the update of future critics. Inspired by this work, we also adopt two critic networks and their target estimates, which are parameterized by $\omega$, $\Omega$, $\omega'$ and $\Omega'$, respectively. Since we plan to deal with our problem through DRL, we can substitute sampled minibatches into the loss functions of action-values (2), which are given by

$$L(\omega) = \frac{1}{N} \sum_{n=1}^{N} (r_n + \gamma Q(s'_n, a'_n|\omega') - Q(s_n, a_n|\omega))^2, \tag{3}$$

$$L(\Omega) = \frac{1}{N} \sum_{n=1}^{N} (r_n + \gamma Q(s'_n, a'_n|\Omega') - Q(s_n, a_n|\Omega))^2, \tag{4}$$

where $a'_n = \mu(s'_n|\theta')$ is the action taken from the target actor network, and $(s_n, a_n, r_n, s'_n)$ is the $n$-th tuple of minibatches stored in replay buffer.

In TD3, the minimum of two critics is employed to serve as the target estimation value. However, updating two Q networks according to the same target estimate will make them less independent, which will further negatively affect the training efficiency. Besides, it is more reasonable to apply the clipped variant of Double Q-Learning to the procedure of actor update. Inspired by the Lagrange relaxation applied in constrained MDP problems Tessler et al. (2018), we propose a dual form of combined value function via a weight factor to determine actions reducing the harmful effect from overestimation. The dual form of combined value function for policy update is given by

$$\pi^* = \arg \min_{0 \leq \lambda \leq 1} \max_{\pi} \left[ (1 - \lambda) Q_\pi(s|\omega) + \lambda Q_\pi(s|\Omega) \right], \forall s \in \chi \setminus \chi'. \tag{5}$$

where $Q_\pi(s|\omega)$ and $Q_\pi(s|\Omega)$ are two critics following policy $\pi$, $\chi'$ be the set of transient states in continuous compact state space $\chi$, and the weight factor $\lambda$ determines the influence of two Q functions. Specifically, (5) will reduce to a normal policy evaluation with one critic when $\lambda = 0$ or $\lambda = 1$. Generally, (5) needs to be solved by a two-timescale approach that may result in a saddle point problem, i.e., on the faster timescale, the policy $\pi$ or its parameter is solved by (5) when fixing $\lambda$, while on the slower timescale, $\lambda$ is slowly increased until the overestimation error is minimized without losing the optimal solution to (5). Due to the potential non-convexity of the action-value function, the method may lead to a sub-optimal solution, and even cause instability when the convergence is slow.

To avoid the potential saddle point problem, we first propose a two-step separation method to solve (5) for each policy update, which determines the optimal action to restrain overestimation based on the target weight factor $\lambda'$ and then use the up-to-date policy to update $\lambda$. Second, although the underestimation bias accompanying the minimization operator is far preferable to overestimation and does not explicitly propagate through the policy update, it indeed negatively affect the policy improvement at every iteration and further brings fluctuation on algorithm convergence. The policy improvement means that the optimized objective function in (5) should steadily increase during training. The insurance of this value improvement lies in the choice of weight factor $\lambda$. The process

of this separation method is given by

$$J(\theta) = \frac{1}{N} \sum_{n=1}^{N} \left[ (1 - \lambda')Q(s_n, a_n|\omega) + \lambda'Q(s_n, a_n|\Omega) \right], \tag{6}$$

$$J(\lambda) = \frac{1}{N} \sum_{n=1}^{N} \left[ (1 - \lambda)Q(s_n, a_n|\omega) + \lambda Q(s_n, a_n|\Omega) \right] * Sign(\lambda), \tag{7}$$

where $0 \leq \lambda' \leq 1$, $a_n = \mu(s_n|\theta)$, which means actions in (6) and (7) should be taken from the policy of current actor instead of the replay buffer. We provide the guarantee of policy improvement by multiplying the averaged return for $\lambda$ updating by a sign function, which is given by

$$Sign(\lambda_i) = \mathbb{I}\left( \frac{1}{N} \sum_{n=1}^{N} \left[ \min_{\lambda_i} \hat{Q}(s_n, a_{n,i}|\omega_i, \Omega_i, \lambda_i) - \hat{Q}(s_n, a_{n,i-1}|\omega_{i-1}, \Omega_{i-1}, \lambda_{i-1}) \right] \right), \tag{8}$$

where

$$\hat{Q}(s, a|\omega, \Omega, \lambda) = (1 - \lambda)Q(s, a|\omega) + \lambda Q(s, a|\Omega), \tag{9}$$

$a_{n,i} = \mu(s_n|\theta_i)$ and $a_{n,i-1} = \mu(s_n|\theta_{i-1})$, which come from actor networks parameterized by current states before and after updating respectively but not from the replay buffer. $\mathbb{I}(x)$ produces 1 when $x$ is negative, and vice versa. The sign denoted in (8) is actually the comparison between the minimum updated Q-values in two critics and the old (before the policy update) combined value defined in (9).

**Lemma 1.** *Denoting the converged values of two critics as $Q(s, a|\omega^\star)$ and $Q(s, a|\Omega^\star)$, respectively, then the convergence of combined value denoted in (9) can be ensured by minimizing (3) and (4).*

Different from updating $\lambda$ in (7), the averaged return for $\theta$ updating in (6) adopts the target weight factor $\lambda'$. Then $(\omega', \Omega', \theta', \lambda')$ are updated adopting the "soft" target updates Lillicrap et al. (2015) by $(\omega, \Omega, \theta, \lambda)$, in the way of

$$\omega_i' \leftarrow \tau \arg\min_{\omega_i} L(\omega_i) + (1 - \tau)\omega_{i-1}',$$
$$\Omega_i' \leftarrow \tau \arg\min_{\Omega_i} L(\Omega_i) + (1 - \tau)\Omega_{i-1}',$$
$$\theta_i' \leftarrow \tau \arg\max_{\theta_i} J(\theta_i) + (1 - \tau)\theta_{i-1}',$$
$$\lambda_i' \leftarrow \tau \arg\min_{\lambda_i} J(\lambda_i) + (1 - \tau)\lambda_{i-1}', \tag{10}$$

where $\tau < 1$ is the factor to control the speed of policy updates for the sake of small value error at each iteration, and $\lambda'$ updates following $\theta'$. We organize the above procedures as the adaptive delayed deep deterministic policy gradient (AD3) algorithm, whose pseudocode is described by Algorithm 1.

**Theorem 1.** *AD3 algorithm asymptotically converges as the iteration $i \to \infty$ with properly chosen learning rate.*

**Theorem 2.** *AD3 algorithm has the property of asymptotical expected policy improvement. Specifically, when the critics tend to converge, i.e., $\exists K, \forall i \geq K, \forall \varepsilon > 0$,*

$$\left| \mathbb{E}_{(s,a)} \left[ \hat{Q}(s, a|\omega_{i+1}, \Omega_{i+1}, \lambda_i) - \hat{Q}(s, a|\omega_i, \Omega_i, \lambda_i) \right] \right| < \varepsilon, \tag{11}$$

*then*

$$\mathbb{E}_{(s,a)} \left[ \hat{Q}(s, a|\omega_{i+1}, \Omega_{i+1}, \lambda_{i+1}) \right] \geq \mathbb{E}_{(s,a)} \left[ \hat{Q}(s, a|\omega_i, \Omega_i, \lambda_i) \right]. \tag{12}$$

The proof of Theorem 1 and Theorem 2 can be found in the Appendix.

---

**Algorithm 1** AD3 Algorithm

---

1: **Input**: The batch size $N$, the maximum of updates $M$, the timeout step $T$, and the soft update parameter $\tau$.
2: **Initialization**: Initialize parameters $(\omega, \Omega, \theta, \lambda) \leftarrow (\omega_0, \Omega_0, \theta_0, \lambda_0)$, $(\omega', \Omega', \theta', \lambda') \leftarrow (\omega_0', \Omega_0', \theta_0', \lambda_0')$ randomly; Initialize replay buffer $R$, the counter $i \leftarrow 0$.
3: **while** $i < M$ **do**
4:     Reset randomly the initial state $s_1$.
5:     **for** $t = 1, T$ **do**
6:         Select action $a_t$ according to the current behavior policy, i.e., $\mu(s_t|\theta_i)$ added by exploration noise;
7:         Execute actions $a_t$, get next states $s_{t+1}$, and immediate reward $r_t$;
8:         Store transition $(s_t, a_t, r_t, s_{t+1})$ in $R$;
9:         **if** $R$ is full **then**
10:           Randomly and uniformly sample the slot $(s_i, a_i, r_i, s_{i+1})$ from $R$;
11:           Minimize the $Q_1$ loss function shown in (3) by gradient decent, and then update $\omega_i$;
12:           Minimize the $Q_2$ loss function shown in (4) by gradient decent, and then update $\Omega_i$;
13:           Maximize the expected return shown in (6) by gradient ascent, and then update $\theta_i$;
14:           Minimize the product of (7) and (8) by gradient ascent, and then update $\lambda_i$;
15:           Execute the "soft" target updates shown in (10) to update $\theta_i'$, $\omega_i'$, $\Omega_i'$, and $\lambda_i'$;
16:           $i \leftarrow i + 1$;
17:         **end if**
18:     **end for**
19: **end while**

---

## UNBIASED ADAPTIVE DELAYED DEEP DETERMINISTIC POLICY GRADIENT

To further improve the performance of proposed AD3 algorithm, we employ AD3 under the framework of unbiased DRL (UDRL) Zhang & Huang (2020). UDRL attempts to solve the systematic bias induced by the approximation of MDP samples. This systematic bias happens in the experience replay mechanism without importance sampling (IS) Precup et al. (2000); Hachiya et al. (2008); Mahmood et al. (2014); Thomas & Brunskill (2016); Jiang & Li (2016); Wang et al. (2016); Foerster et al. (2017); Metelli et al. (2018), because there exists mismatch between distributions of the target policy and the behavior policy. Without IS to weight the tuples with different probabilities in commonly applied off-policy methods, the experience replay that memorizes the past observations for random samples will accumulate the systematic errors and lower the convergence performance. When applying UDRL, the independently and identically distributed (IID) initial states are parallelly sampled to start respective tuples at the beginning of each iteration. Then the parallel virtual agents follow the same behavior policy to complete their tuples, which serve as the observations to synchronously train and update the shared network parameters.

In the case of unbiased AD3 (UAD3) method, the parallelly sampled IID observations should be used to train the two critic networks, an actor network and a weight factor. At each iteration, the actions are taken following the same behavior to receive rewards and next states, so that the achieved four-tuple transition slots are independent and follow the same joint probability. By this means, no IS is required in the approximations of (3), (4), (6), (7) and (8). The pseudocode of UAD3 is organized as Algorithm 2.

## EXPERIMENTS

### CONTINUOUS MAZE

One of the benchmark tasks we choose is the continuous maze which is filled with obstacles. The environment of the continuous maze problem includes infinite states and actions which is shown in Fig. 1(a). At every step, the agent is able to move towards all directions with its step size. Since the state-action space is continuous, the agent may travel through the obstacles represented by the gray grids if no effective guide is provided during the whole training. The dark solid line at the edge of Fig. 1(a) is represented as the wall to show the maze is closed except for the goal. The task of

---

**Algorithm 2** UAD3 Algorithm

---

1: **Input**: The batch size $N$, the maximum of updates $M$, and the soft update parameter $\tau$.
2: **Initialization**: Initialize parameters $(\omega, \Omega, \theta, \lambda) \leftarrow (\omega_0, \Omega_0, \theta_0, \lambda_0)$, $(\omega', \Omega', \theta', \lambda') \leftarrow (\omega'_0, \Omega'_0, \theta'_0, \lambda'_0)$ randomly.
3: **for** $i = 1, M$ **do**
4:     Sample $S_i = (s_{i,1}, s_{i,2}, \cdots, s_{i,N})$ IID;
5:     Choose actions $A_i = (a_{i,1}, a_{i,2}, \cdots, a_{i,N})$ for $S_i$ according to the current actor network $\mu(S_i|\theta_i)$ added by exploration noise;
6:     Execute actions $A_i$, get next states $S'_i = (s'_{i,1}, s'_{i,2}, \cdots, s'_{i,N})$ and immediate rewards $R_i = (r_{i,1}, r_{i,2}, \cdots, r_{i,N})$;
7:     Minimize the $Q_1$ loss function shown in (3) by gradient decent, and then update $\omega_i$;
8:     Minimize the $Q_2$ loss function shown in (4) by gradient decent, and then update $\Omega_i$;
9:     Maximize the expected return shown in (6) by gradient ascent, and then update $\theta_i$;
10:    Minimize the product of (7) and (8) by gradient ascent, and then update $\lambda_i$;
11:    Execute the "soft" target updates shown in (10) to update $\theta'_i$, $\omega'_i$, $\Omega'_i$, and $\lambda'_i$;
12: **end for**

---

this experiment is move the agent from the *start* to the *goal* colored yellow with no block. This goal can be achieved by setting scores for the agent at every step. Specifically, the agent receives negative if it encounters any jam. If the agent reaches the *goal*, it will be rewarded $100$ score. In other blank areas, the reward is set as the minus distance from the agent to the *goal* for the purpose of stimulating the agent to fulfill the task as soon as possible.

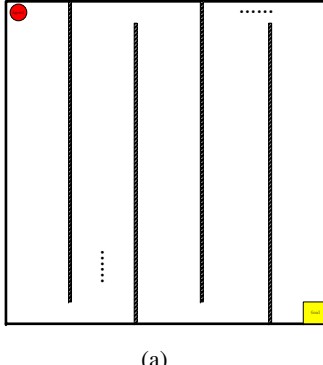
(a)

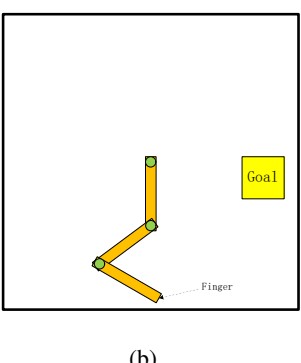
(b)

Figure 1: (a) The maze environment with continuous state-action space and lines of obstacles; (b) The robot arm environment with a move and grasp task.

In this experiment, we evaluate the performance of AD3 and UAD3 algorithms, using the baseline algorithms DDPG and TD3. The hyperparameters are shown in Table 1. Every 500 iterations (update periods), an evaluation procedure is launched, which records 100 episodes and averages their results to improve accuracy, i.e., the average reward, where each episode observes the agent from the *start* to the *goal* and adds up the rewards without discount along the path. Similar evaluation procedures are shared by all experiments in the context with the same cycle.

Figure 2 illustrates the average reward versus update periods of the continuous maze with barriers of different lines plotted in Fig. 1(a). Fig. 2(a) shows that AD3 converges faster than DDPG and TD3. From Figs. 2(b)-2(c), we see UAD3 robustly converges so that the agent can reach the goal and receives a positive reward, however, other algorithms diverges and fails in their missions. The better performance is mainly due to the policy improvement clarifies in Theorem 2. Besides, adaptive $\lambda$ provides a superior way to reduce overestimation.

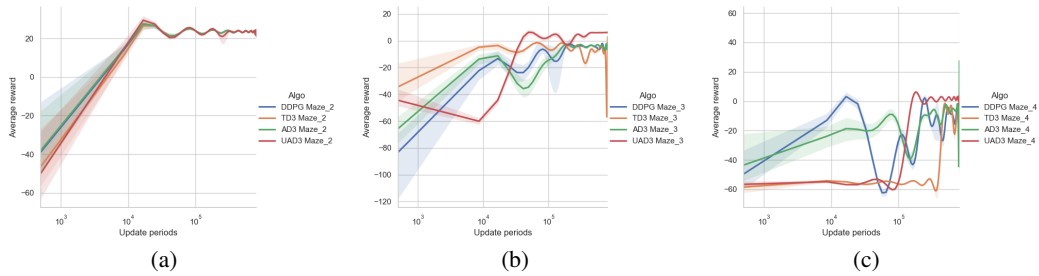

Figure 2: Convergence performance of the continuous maze with barriers (a) of 1 line; (b) of 2 lines; (c) of 3 lines.

ROBOT ARM

The "robot arm" experiment is a move and grasp task which is shown in Fig. 1(b). In this figure, we just show three sections to represent a general arm that may contain several more sections. The aim of this task is to move the $finger$ represented by the end of the arm to catch the $goal$. Specifically, the $finger$ should get into the yellow "goal box", and then hold on to the moving goal for required steps to fulfill the grasp task. In the experiment setting, the $goal$ is randomly moving, so the state representation includes both the positions of joints represented by the green circles and their relative positions to the goal. Besides, the number of sections determines the dimension of action space which contains the rotation angle of each section. The reward is set as the negative distance from the $finger$ to the $goal$ outside the "goal box". If the $finger$ is located within the "goal box", the reward is set as 1.

We also evaluate our proposed algorithms using the baseline algorithms DDPG and TD3 based on the hyperparameters in Table 1. Fig. 3 shows the computational performance of the robot arm with $2-7$ sections by fitting scatterplots run for 800 thousand iterations. Notably, one more section will raise the state dimension by 4, including the 2-dimensional coordinates of joints and their relative coordinates to the goal. Increased state dimension needs more time for convergence and produces lower converged average reward. Throughout Fig. 3, we observe UAD3 can robustly converge to a value much higher than other algorithms, which shows the best performance because higher converged average reward means the agent can react more promptly to the moving goal. From Figs. 3(a)-3(b), AD3 converges faster and more robustly to a higher value compared with DDPG and TD3 under the same circumstance. In Fig. 3(c), AD3 converges above the zero line with a value higher than DDPG for 6 sections, however, TD3 diverges for both 6 and 7 sections. Overall, UAD3 and AD3 have better performance than their counterparts DDPG and TD3.

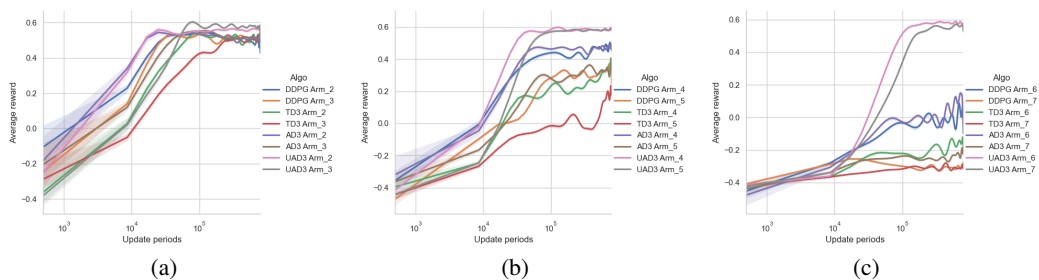

Figure 3: Convergence performance of the robot arm with (a) 2-3 sections; (b) 4-5 sections; (c) 6-7 sections.

CLASSICAL CONTROL ENVIRONMENT

In this part, we adopt a series of classical control experiments including Pendulum, Acrobat, continuous mountain car and Cartpole as the benchmark tasks to evaluate the proposed AD3 and UAD3 algorithms based on the hyperparameters in Table 1.

It is noteworthy that we revised the rewards of Cartpole environment to make them more challenging. Specifically, the distance and average velocity of the cart are added to the rewards to stimulate the cart to go farther and faster because staying at the origin trading for stability is not enough to tell the robustness of algorithms. The results of the average reward, average velocity and distance versus update periods for Cartpole are given by Fig. 4. From Fig. 4(a), we see UAD3 and AD3 can converge much faster and more stably than DDPG and TD3. Besides, UAD3 and AD3 are able to simulate the cart to move faster and farther according to Figs. 4(b) and 4(c), respectively. Figs. 5(a)-5(c) present the results of average reward versus update periods for Pendulum, Acrobat and continuous mountain car, which further show the advantages of UAD3 and AD3 over DDPG and TD3. Moreover, we reproduce the results of UDDPG Zhang & Huang (2020) for a fair comparison with UAD3. From Figs. 4 and 5, we see that UAD3 is even better than UDDPG in higher convergence speed, converged average reward, and stability.

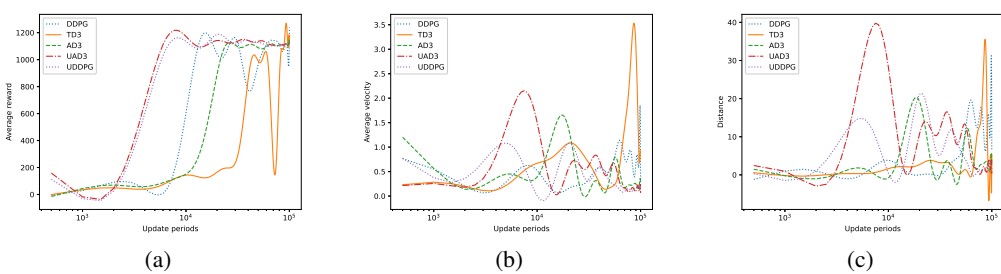

Figure 4: (a) Average reward; (b) Average velocity; (c) Distance versus update periods in Cartpole.

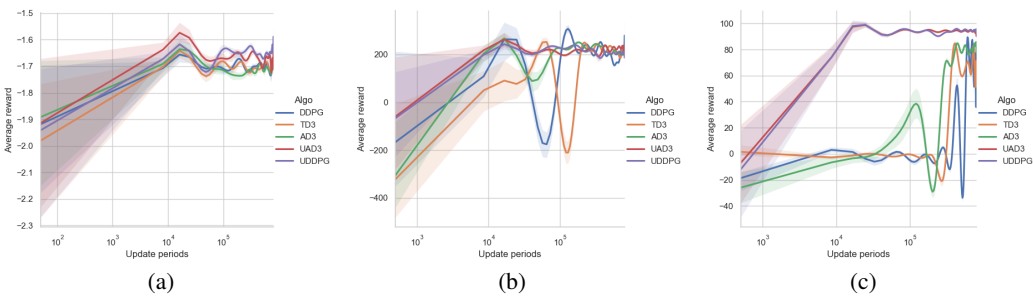

Figure 5: Computational efficiency in (a) in Pendulum; (b) Acrobat; (c) continuous mountain car.

CONCLUSION

In this paper, we proposed the combined value of two independent critics connected by a weight factor to update the policy, which will in turn to update the combined value. We also proposed the objective function for updating the weight factor by multiplying the updated combined value by a sign, which compares the minimum updated Q-value in two critics with the combined value before updating. Based on the above two components, we present AD3 to reduce the overestimation bias and ensure future policy improvement at the same time. Furthermore, we apply AD3 to UDRL framework to eliminate the systematic bias caused by the probability mismatch between the behavior policy and the target policy in experience replay, and present UAD3. The proposed AD3 algorithm is theoretically proved to possess the property of asymptotical convergence and expected policy

improvement. Evaluation results show that our proposed algorithms can boost and stabilize the convergence. Although we represent the weight factor as a variable in the context, it can be formulated as a function of states. It can be seen that all theorems and proofs can apply to it when lambda is state-dependent, and all the experimental results are based on the model of state-dependent weight factor $\lambda(s)$. The network architecture of weight factor concerning states is clarified in Appendix.

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

## APPENDIX

### PROOF OF LEMMA 1

*Proof*

$$\mathbb{E}_{(s,a)} \left[ (\hat{Q}(s,a|\omega, \Omega, \lambda) - \hat{Q}(s,a|\omega^\star, \Omega^\star, \lambda))^2 \right],$$
$$= \mathbb{E}_{(s,a)} \left[ ((1-\lambda)(Q(s,a|\omega) - Q(s,a|\omega^\star)) + \lambda(Q(s,a|\Omega) - Q(s,a|\Omega^\star)))^2 \right],$$
$$\leq \mathbb{E}_{(s,a)} \left\{ [(1-\lambda)(Q(s,a|\omega) - Q(s,a|\omega^\star))]^2 + [\lambda(Q(s,a|\Omega) - Q(s,a|\Omega^\star))]^2 \right\},$$
$$\to 0, \tag{13}$$

where the converged combined value $\hat{Q}(s,a|\omega^\star, \Omega^\star, \lambda) = (1-\lambda)Q(s,a|\omega^\star) + \lambda Q(s,a|\Omega^\star)$. $\qquad\square$

### PROOF OF THEOREM 1

*Proof*    This proof is based on Lemma 1 of SINGH et al. (2000), which is moved originally below for convenience.

Lemma 1 of SINGH et al. (2000): Consider a stochastic process $(\alpha_t, \Delta_t, F_t)$, $t \geq 0$, where $\alpha_t, \Delta_t, F_t : X \to \mathbb{R}$, which satisfies the equations

$$\Delta_{t+1}(x) = (1 - \alpha_t(x))\Delta_t(x) + \alpha_t(x)F_t(x), \quad x \in X, t = 0, 1, 2, \cdots \tag{14}$$

Let $P_t$ be a sequence of increasing $\sigma$-fields such that $\alpha_0, \Delta_0$ are $P_0$-measurable and $\alpha_t, \Delta_t$ and $F_{t-1}$ are $P_t$-measurable, $t = 1, 2, \cdots$ Assume that the following hold:

1. the set of possible states X is finite.
2. $0 \leq \alpha_t(x) \leq 1, \sum_t \alpha_t(x) = \infty, \sum_t \alpha_t^2(x) < \infty$ $w.p.1$.
3. $\|\mathbb{E}\{F_t(\cdot)|P_t\}\| \leq \gamma\|\Delta_t\| + c_t$, where $\gamma \in [0, 1)$ and $c_t$ converges to zero $w.p.1$.
4. $Var\{F_t(x)|P_t\} \leq K(1 + \|\Delta_t\|)^2$, where $K$ is some constant.

Then $\Delta_t$ converges to zero with probability one $(w.p.1)$.

Within the scope of this paper, the MDP state space is finite, satisfying condition 1 in Lemma 1 of SINGH et al. (2000), and Lemma condition 2 holds by proper selection of learning rate. According to Szepesvári (2010), even the commonly used constant learning rate can make algorithms converge in distribution.

We apply Lemma 1 of SINGH et al. (2000) with $P_t = \{Q(\cdot, \cdot|\omega_0), Q(\cdot, \cdot|\Omega_0), s_0, a_0, r_1, s_1, \cdots, s_t, a_t\}$. Following the update rule for optimizing (3) and (4) and using the current policy to produce the action $a_{t+1} = \mu(s_{t+1}|\theta)$, we have

$$Q(s_t, a_t|\omega_{t+1}) = (1 - \alpha_t)Q(s_t, a_t|\omega_t) + \alpha_t [r_t + \gamma Q(s_{t+1}, a_{t+1}|\omega_t)], \tag{15}$$
$$Q(s_t, a_t|\Omega_{t+1}) = (1 - \alpha_t)Q(s_t, a_t|\Omega_t) + \alpha_t [r_t + \gamma Q(s_{t+1}, a_{t+1}|\Omega_t)]. \tag{16}$$

Under the setting of our proposed algorithm, we denote $\Delta_t = \hat{Q}(\cdot, \cdot|\omega_t, \Omega_t, \lambda) - Q^\star(\cdot, \cdot)$, which is the difference between the combined value of weighted critics denoted in (9) and optimal value function. Then we have

$$\Delta_{t+1}(s_t, a_t) = \hat{Q}(s_t, a_t|\omega_{t+1}, \Omega_{t+1}, \lambda) - Q^\star(s_t, a_t),$$
$$= (1-\lambda)Q(s_t, a_t|\omega_{t+1}) + \lambda Q(s_t, a_t|\Omega_{t+1}) - Q^\star(s_t, a_t),$$
$$= (1-\alpha_t) \left[ \hat{Q}(s_t, a_t|\omega_t, \Omega_t, \lambda) - Q^\star(s_t, a_t) \right] + \alpha_t F_t,$$
$$= (1-\alpha_t)\Delta_t(s_t, a_t) + \alpha_t F_t(s_t, a_t), \tag{17}$$

where the third equality is due to the substitution of (15) and (16), and

$$F_t(s_t, a_t) = r_t + \gamma \left[ \hat{Q}(s_{t+1}, a_{t+1} | \omega_t, \Omega_t, \lambda) \right] - Q^\star(s_t, a_t). \tag{18}$$

Since the reward is bounded within the scope of this paper, the action-values are also bounded, then condition 4 in Lemma 1 of SINGH et al. (2000) holds. According to the proof in Theorem 2 of SINGH et al. (2000), there is $\| \mathbb{E} \left[ F_t(s_t, a_t) | P_t \right] \| \le \gamma \|\Delta_t\|$, which satisfies condition 3 in Lemma 1 of SINGH et al. (2000).

Finally, it can be concluded that $\hat{Q}(\cdot, \cdot | \omega_t, \Omega_t, \lambda)$ converges to $Q^\star(\cdot, \cdot)$ with probability 1.

$\square$

PROOF OF THEOREM 2

*Proof*    If $sign_{i+1} \ge 0$, then

$$\begin{aligned}
&\mathbb{E}_{(s,a)} \left[ \hat{Q}(s, a | \omega_{i+1}, \Omega_{i+1}, \lambda_{i+1}) \right] \\
&= \mathbb{E}_s \left[ \hat{Q}(s, \mu(s|\theta_{i+1}) | \omega_{i+1}, \Omega_{i+1}, \lambda_{i+1}) \right] \\
&\approx \frac{1}{N} \sum_{n=1}^{N} \hat{Q}(s_n, \mu(s_n|\theta_{i+1}) | \omega_{i+1}, \Omega_{i+1}, \lambda_{i+1}) \\
&\ge \frac{1}{N} \min \left( \sum_{n=1}^{N} Q(s_n, \mu(s_n|\theta_{i+1}) | \omega_{i+1}), \sum_{n=1}^{N} Q(s_n, \mu(s_n|\theta_{i+1}) | \Omega_{i+1}) \right) \\
&\ge \frac{1}{N} \sum_{n=1}^{N} \left[ \hat{Q}(s_n, \mu(s_n|\theta_i) | \omega_i, \Omega_i, \lambda_i) \right] \\
&\approx \mathbb{E}_{(s,a)} \left[ \hat{Q}(s, a | \omega_i, \Omega_i, \lambda_i) \right],
\end{aligned} \tag{19}$$

where the approximations the mean average is statistically equal to the expectation (or unbiased approximation for UAD3), the first inequality holds based on (9), and the second inequality holds because (8) is no less than 0.

Otherwise, if $sign_{i+1} \le 0$, then

$$\begin{aligned}
&\mathbb{E}_{(s,a)} \left[ \hat{Q}(s, a | \omega_{i+1}, \Omega_{i+1}, \lambda_{i+1}) \right] \\
&= \mathbb{E}_s \left[ \hat{Q}(s, \mu(s|\theta_{i+1}) | \omega_{i+1}, \Omega_{i+1}, \lambda_{i+1}) \right] \\
&\approx \frac{1}{N} \sum_{n=1}^{N} \hat{Q}(s_n, \mu(s_n|\theta_{i+1}) | \omega_{i+1}, \Omega_{i+1}, \lambda_{i+1}) \\
&\ge \frac{1}{N} \sum_{n=1}^{N} \hat{Q}(s_n, \mu(s_n|\theta_{i+1}) | \omega_{i+1}, \Omega_{i+1}, \lambda_i) \\
&= \frac{1}{N} \sum_{n=1}^{N} \hat{Q}(s_n, \mu(s_n|\theta_{i+1}) | \omega_{i+1}, \Omega_{i+1}, \lambda_i') \\
&\ge \frac{1}{N} \sum_{n=1}^{N} \hat{Q}(s_n, \mu(s_n|\theta_i) | \omega_{i+1}, \Omega_{i+1}, \lambda_i') \\
&= \frac{1}{N} \sum_{n=1}^{N} \hat{Q}(s_n, \mu(s_n|\theta_i) | \omega_{i+1}, \Omega_{i+1}, \lambda_i) \\
&\approx \mathbb{E}_s \left[ \hat{Q}(s, \mu(s|\theta_i) | \omega_i, \Omega_i, \lambda_i) \right] \\
&= \mathbb{E}_{(s,a)} \left[ \hat{Q}(s, a | \omega_i, \Omega_i, \lambda_i) \right],
\end{aligned} \tag{20}$$

where the first inequality holds due to the fact that the update of $\lambda$ is to maximize (7) given a negative sign, the second inequality holds because the update of $\theta$ is to maximize (6). Although (19) and (20) are done for immediate target updates, the same conclusions can be achieved under the condition of delayed or "soft" target updates if the Q network is linear with respect to the actor and critic parameters.    $\square$

NETWORK ARCHITECTURE

We construct the critic network using a fully-connected MLP with two hidden layers. The input is composed of the state and action, outputting a value representing the Q-value. The ReLU function is adopted to activate the first hidden layer. The setting of actor network is similar to that of the critic network, except that the input is the state and the output is multiplied by the action supremum after tanh nonlinearity. The network of weight factor $\lambda$ is constructed similar to the actor network except replacing the tanh nonlinearity by clipping $\lambda$ in $[0, 1]$. The architecture of networks are plotted in Fig. 6.

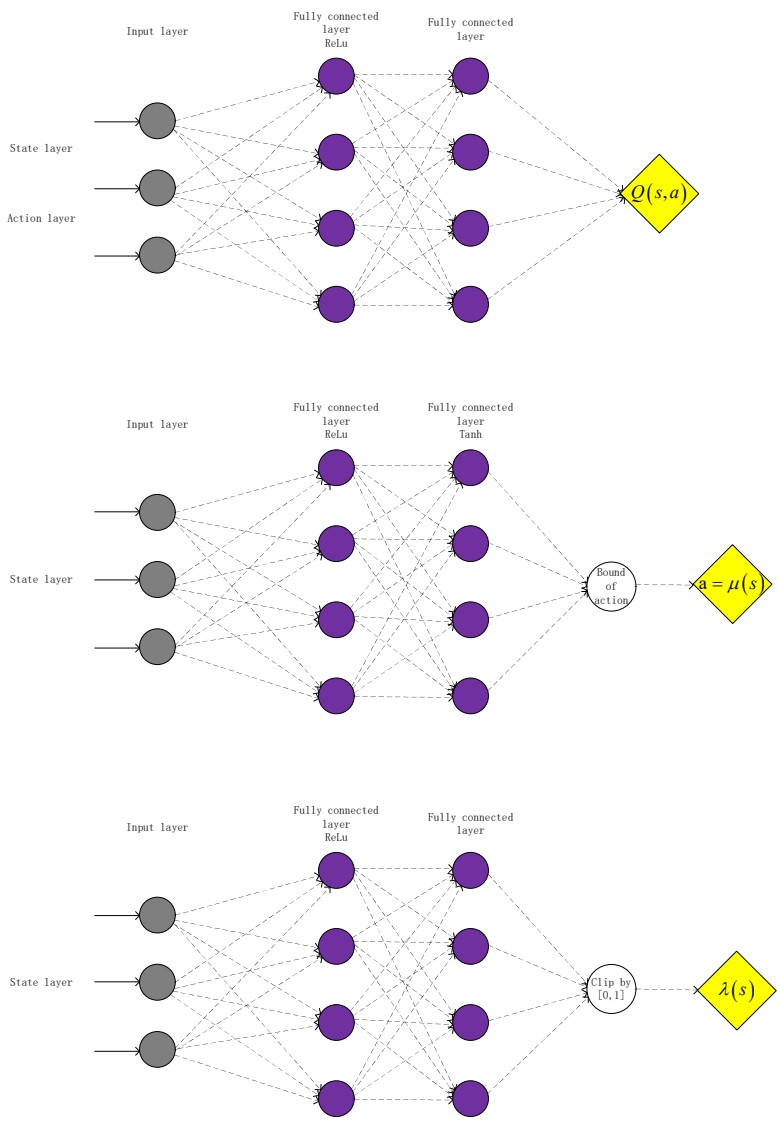

Figure 6: Architecture of networks.

HYPERPARAMETERS

Table 1 lists the common hyperparameters shared by all experiments and their respective settings.

Table 1: **List of hyperparameters**

| Shared Env | Value | Description | Algorithm applied |
|---|---|---|---|
| $LR\_a$ | 0.001 | Learning rate of actor | DDPG, TD3, AD3, UAD3, UDDPG |
| $LR\_c$ | 0.001 | Learning rate of critic | DDPG, UDDPG |
| $LR\_c_1$ | 0.001 | Learning rate of critic1 | TD3, AD3, UAD3 |
| $LR\_c_2$ | 0.001 | Learning rate of critic2 | TD3, AD3, UAD3 |
| $LR\_\lambda$ | 0.001 | Learning rate of weight factor | TD3, AD3, UAD3 |
| $\tau\_a$ | 0.01 | Soft update parameter of actor | DDPG, UDDPG |
| $\tau\_c_1$ | 0.01 | Soft update parameter of critic1 | TD3, AD3, UAD3 |
| $\tau\_c_2$ | 0.01 | Soft update parameter of critic2 | TD3, AD3, UAD3 |
| $\gamma$ | 0.9 | Discount horizon factor | DDPG, TD3, AD3, UAD3, UDDPG |
| Interval | 500 | Eval period | DDPG, TD3, AD3, UAD3, UDDPG |
| Test | 100 | Episodes per eval period | DDPG, TD3, AD3, UAD3, UDDPG |
| Var_dr | 0.9995 | Exploration variance decay rate | DDPG, TD3, AD3, UAD3, UDDPG |
| Sample | 200 | Sample size of initial states | UAD3, UDDPG |
| Batch | 200 | Size of mini-batches | DDPG, TD3, AD3 |
| **Cartpole** | **Value** | **Description** | **Algorithm applied** |
| Max_EPS | 500 | Maximal steps per episode training | DDPG, TD3, AD3 |
| Runout | 1000 | Maximal steps per episode eval | DDPG, TD3, AD3, UAD3, UDDPG |
| Initial variance | 10.0 | Initial exploration variance | DDPG, TD3, AD3, UAD3, UDDPG |
| Memory | 50000 | Size of replay buffer | DDPG, TD3, AD3 |
| Train_num | 100000 | Updating iterations (Not episodes) | DDPG, TD3, AD3, UAD3, UDDPG |
| **Acrobot MountainCar** | **Value** | **Description** | **Algorithm applied** |
| Max_EPS | 500 | Maximal steps per episode training | DDPG, TD3, AD3 |
| Runout | 500 | Maximal steps per episode eval | DDPG, TD3, AD3, UAD3, UDDPG |
| Initial variance | 10.0 | Initial exploration variance | DDPG, TD3, AD3, UAD3, UDDPG |
| Memory | 10000 | Size of replay buffer | DDPG, TD3, AD3 |
| Train_num | 800000 | Updating iterations (Not episodes) | DDPG, TD3, AD3, UAD3, UDDPG |
| **Robot Arm** | **Value** | **Description** | **Algorithm applied** |
| Max_EPS | 200 | Maximal steps per episode training | DDPG, TD3, AD3 |
| Runout | 100 | Maximal steps per episode eval | DDPG, TD3, AD3, UAD3, UDDPG |
| Initial variance | 1.0 | Initial exploration variance | DDPG, TD3, AD3, UAD3, UDDPG |
| Memory | 30000 | Size of replay buffer | DDPG, TD3, AD3 |
| Train_num | 800000 | Updating iterations (Not episodes) | DDPG, TD3, AD3, UAD3, UDDPG |
| **Maze** | **Value** | **Description** | **Algorithm applied** |
| Max_EPS | 500 | Maximal steps per episode training | DDPG, TD3, AD3 |
| Runout | 100 | Maximal steps per episode eval | DDPG, TD3, AD3, UAD3, UDDPG |
| Initial variance | 10.0 | Initial exploration variance | DDPG, TD3, AD3, UAD3, UDDPG |
| Memory | 10000 | Size of replay buffer | DDPG, TD3, AD3 |
| Train_num | 800000 | Updating iterations (Not episodes) | DDPG, TD3, AD3, UAD3, UDDPG |
| **Pendulum** | **Value** | **Description** | **Algorithm applied** |
| Max_EPS | 200 | Maximal steps per episode training | DDPG, TD3, AD3 |
| Runout | 100 | Maximal steps per episode eval | DDPG, TD3, AD3, UAD3, UDDPG |
| Initial variance | 10.0 | Initial exploration variance | DDPG, TD3, AD3, UAD3, UDDPG |
| Memory | 30000 | Size of replay buffer | DDPG, TD3, AD3 |
| Train_num | 800000 | Updating iterations (Not episodes) | DDPG, TD3, AD3, UAD3, UDDPG |

ESTIMATE OF Q-VALUE

We uniformly sample 10000 states from the replay buffer every 500 update periods, and average the computed Q-values based on the newly-updated current policy. The averaged Q-value can be seen as the estimate of discounted cumulative return (value estimate). The value estimates of Cartpole, Robotarm of 2 sections and Pendulum experiments are plotted in Fig. 7. From these figures, we can see the converged value function $Q(s, a|\omega^\star)$ denoted in Lemma 1.

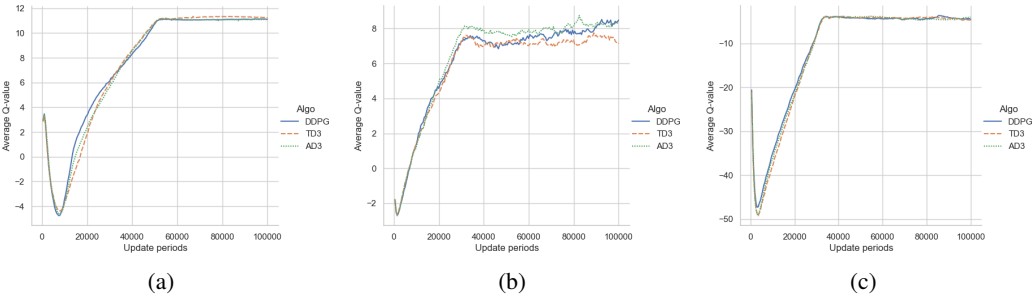

Figure 7: Value estimate in (a) Cartpole; (b) Robotarm of 2 sections; (c) Pendulum.

