# OpenReview forum: "Deep Reinforcement Learning With Adaptive Combined Critics"
_ICLR.cc/2021/Conference — Reject_

### Official Review · AnonReviewer2 · 2020-10-17
**Unsure about correctness and insufficient experimental evaluation**

**Rating:** 3
**Confidence:** 4

**Review:**

Summary:

The paper proposes a method to avoid the overestimation bias. They modify the TD3 algorithm such that, instead of regressing the two Q-values towards the minimum of both Q-value functions, the regression target is a convex combination of both, determined by a parameter lambda which is updated on a slower time-scale.

I have three concerns due to which I am currently recommending rejection: First, the smaller concern is that the paper is currently difficult to understand, second, my major concern is that I am not sure about it's correctness, due to equation (9). Lastly, the experimental evaluation is insufficient.

Regarding understanding: The paper currently has a significant number of grammatical mistakes and unusual or wrong word choices. While this often is without consequence, unfortunately, at times it does impact the clarity and creates uncertainty about the argument the authors are trying to make. However, overall I believe I was able to understand the algorithm the authors are proposing, so this is not my major concern.

Regarding correctness of the algorithm: My main issue is with equation (9), i.e. with how lambda (the trade-off parameter in the convex combination between the two q-value functions) is updated. I believe equation (9) should almost always be negative because the minimum of two values is always smaller or equal than any convex combination of both, since lambda is restricted to 0 ≤ lambda ≤ 1. Which means that based on equation (11), lambda is actually optimized to *maximize* the value, i.e. it encourages overestimation instead of preventing it.

One caveat might be that equation (9) actually compares Q-values from before and after each update. It is not explained why this is done, but I believe that should  change the sign of (9) in only very rare cases as I'd expect Q to move only very little in each update.

Lastly, lambda is a global variable, whereas the minimum operation of TD3 is local, i.e. performed independently in each state. I believe this is a big difference in the proposed algorithms which is not discussed at all.

Regarding experimental evaluation: As the authors propose a novel, general purpose algorithm which aims to improve on TD3, it is important to evaluate it on a wide range of benchmarks, including the typical, widely used mujoco environments on which most current continuous control algorithms have been evaluated and compared.

Other, minor, points:

- In equation (5): What is the min over? What is the max over? Also, the Q functions have the wrong argument signature
- Below equation (5): I don't understand what xi and xi' are
- Very minor point for Equation (6) & (7): Personally I don't like the introduction of the \bar{} version of lambda and would find (1-lambda) clearer in the equation

---

### Official Review · AnonReviewer3 · 2020-10-27

**Rating:** 3
**Confidence:** 4

**Review:**

The authors propose a deterministic policy-gradient algorithm that extends the TD3 algorithm (Fujimoto 2018). The main claim is that it reduces overestimation issues in a more effective way. Two Q-critics are maintained with separate parameters, but updated using the same transitions. Then a convex combination of these critics is used in the deterministic policy gradient update. The mixture parameter is learned on a slower time-scale to minimize this convex combination over states (instead of taking the minimum of the 2 critics per batch as in TD3). Another contribution in the paper is the “Unbiased” variant of the algorithm (UAD3), which addresses the off-policy nature of the replay mechanism of the AD3 algorithm described above. My understanding is that this is simply a version of the algorithm that does not use any replay mechanism and samples the state iid from the on-policy distribution, so it isn’t a novel idea in itself.

There are two theorems given to justify the algorithm choices, but I want to question their validity. The first one says that AD3 converges asymptotically, but no formal statement of what this means is given and the proof for it in the appendix only states broad facts about stochastic approximation, but nothing specific that applies to the AD3 algorithm. Theorem 2 is misleading in another way, it says that AD3 has “the property of asymptotical expected policy improvement”, but it only really says that the critic value will be increasing, not that the actual policy value is increasing (and so an actual policy improvement step). Moreover, the proof contains some approximation steps which are not justified.

The approach is tested in two simple continuous control environments (maze + reacher task). (Are these using the full state as input?). There the proposed approaches perform better it seems than the baselines (TD3 and DDPG), but there isn’t any analysis to understand whether that was due to better critics - why not plot the estimated and true returns during learning to see whether AD3 indeed does better than the other critic update strategies? The experimental section is missing details to make these results reproducible and interpretable, for example what network architecture was used for the policy and critic?  All learning curves have rather strange oscillation patterns. Is that an artifact of the smoothing used? How many seeds were used to obtain each learning curve?

At the moment, the advantages of the proposed approach are neither demonstrated theoretically or empirically in a satisfactory way (see comments above). At least one of these aspects need to be improved significantly before the paper is ready for acceptance.

Other comments:

The objective to decide how to mix the critics could be better motivated. Making sure that the critic increase may not be the best criterion to ensure stability and reduce over/underestimation for example.

Minor things:

* “In actor-critic methods, the policy is deterministic…” Actor-critic methods are actually more commonly described in a stochastic policy setting I believe. This work seems to be based on the deterministic policy gradient paper which is not cited.
* Citation style should use parentheses around names in most cases in the text.
* Gattami 2019 is not the right reference for the Bellman equations and derived RL updates.
* Lots of equations are redundant and add unnecessary complexity. For example Eq 3 and Eq 4 are the same equations with different parameters, so why not call the two parameters w_1 and w_2 and write the equation once with w_i  for i \in {1,2}
* The hat on the \lambda in Eq 6,7 are hard to see because they are attached to the left bracket.
* adopts -> adopt
* Transition slots -> tuples?
* Minus distance/reward -> negative

---

### Official Review · AnonReviewer4 · 2020-10-28
**I don’t think the proposed method mitigates the overestimation bias directly.**

**Rating:** 5
**Confidence:** 4

**Review:**

This paper proposes new value-based deep reinforcement learning algorithms (AD3 and UAD3) to address the overestimation bias issue of Q learning. The main contributions of this paper are three folds: 1) The authors propose a weighted sum of two state-action value functions that are trained separately. Then, it is used to update the policy. 2) The mixing weights are updated by the two-step separation method. 3) The original method (AD3) is integrated with the idea of unbiased DRL (Zhang and Huang, 2020).
I have the following concerns for this paper:
1) Although the algorithms are described clearly, I do not fully understand why the proposed method mitigates the overestimation bias issues. The authors claim that applying the clip variant of DDQN to the update procedure of the actor is more reasonable, but the experimental results did not support the authors’ claim. Is it possible to show value estimates by TD3 and the proposed methods?
2) I agree that the two timescale approach may suffer from a saddle point problem when used to solve Equation (5). However, I am not sure whether the two-step separation method can avoid the saddle point problem. Does AD3 outperform the two-timescale approach?
3) The framework of unbiased DRL (UDRL) is interesting, but it is not the main contribution of the paper. So, a comparison of UAD3 with DDPG and TD3 is not fair. Is it possible to apply UDRL to DDPG and TD3? Since Zhang and Huang (2020) proposed Unbiased DDPG, I think the comparison of UAD3 with UDDPG helps understand how AD3 contributes to learning efficiency.
4) The authors compare the proposed methods with DDPG and TD3. However, DDPG does not consider the overestimation issue. It would be better to compare the proposed method with Averaged-DQN (Anschel et al., ICML 2017) and MaxMin Q-learning (Lan et al., ICLR 2020).
5) Is Equation (5) correct? I think - \lambda Q_\pi^2 should be + \lambda Q_\pi^2. In addition, Q_\pi^1 and Q_\pi^2 are not defined explicitly.
Overall, I think the idea of the paper is novel, but further improvements should be added to increase the score of the paper.

---

### Official Review · AnonReviewer1 · 2020-10-28
**The paper presents a potentially interesting method to mitigate the overestimation issue. But it is far away from publication at ICLR.**

**Rating:** 3
**Confidence:** 5

**Review:**

The paper presents an approach to mitigate the overestimation issue, which is quite common in RL algorithms whenever computing boostrap target is needed. The key idea is to introduce a weight parameter between two Q values and adopt a dual problem formulation to learn this weight and the policy parameters. The authors propose a two-step method to estimate these parameters and present some experiments to show the proposed algorithm's performance.

Overall Quality/Clarity is poor. The reasons are as follows.

First, the theorems are incorrect or meaningless. Theorem 1 says the proposed algorithm AD3 (i.e. adaptive delayed deep deterministic policy gradient) converges. But the proof is for the tabular case and it becomes trivial. Furthermore, the proof itself seems to be incorrect. There should be restrictions on the function \mu as it would affect what type of operator it is. Theorem 2 is simply incorrect. The inequality (13) does not imply policy improvement at all. I guess the authors want to say the policy under the policy parameter of the i+1th step has a higher expected return. But the notations are not saying that. The proof itself uses approximation signs arbitrarily, and it is unclear what the meaning is.

Second, experiments lack details and it is unclear what message they are trying to convey (e.x. Fig 2). How were those hyperparameters chosen? How many runs? What is the standard deviation across runs?

Third, many of the statements and notations are not rigorous.

The authors never clearly state the objective function they are using. Is it the same as DDPG? And, it never makes a clear distinction between the objective used for updating the critic parameter and the actor parameter.

The definition of J(\theta) is confusing, what is the underlying distribution of s (first paragraph on page 3)? In eq (6) and (7), the definition of J(\theta) changes.

“In actor-critic methods, the policy is deterministic and commonly parameterized as the actor network.” This is inaccurate. The policy does not have to be deterministic in actor-critic methods.

“However, updating two Q networks according to the same target estimate will make them less independent, which will further negatively affect the training efficiency.” Why?

“Although the underestimation bias accompanying the minimization operator is far preferable ..., it indeed negatively affect the policy improvement at every iteration and further brings fluctuation on algorithm convergence.” Why does it "indeed affect the policy improvement"?

Important detail missing in the background. The authors should cite the DDPG or the DPG papers at least when describing the updating rule (2). In the original work, there is also a slower moving network for the actor used for computing boostrap target to update the critic. For a paper concerning removing overestimation bias, it is a very important detail and should be clearly stated.

Last, those important derivations (i.e. those from eq (8))  do not have justification.

Originality/Significance. I did not see exactly the same work before. But there are similar works and many papers in the same research line are not appropriately discussed. Please see the citation part below.

Lack of citation & unrigorous citation.

"The foundation for updates of network parameters is the Bellman equation Gattami (2019)." The cited paper is about multi-objective and constraint MDP; the credit of the Bellman equation should not go to this work. There is no need to cite anything here. Rigorously, Q-learning is based on the Bellman optimality equation.

Maxmin Q-learning: Controlling the Estimation Bias of Q-learning by Lan et al. This work introduces a method that reduces bias to almost zero while also reducing variance. They also provide convergence proof under the tabular case. Since their approach does not have the issues as the author mentioned for DDQN, the authors should include such a citation and further discuss why their proposed approach is still interesting. The authors should also search for average Q-learning, ensemble Q-learning, etc.

The earliest work discussing the source overestimation should be Issues in Using Function Approximation for Reinforcement Learning by Sebastian Thrun and Anton Schwartz. 1993.

The following nice papers also discuss the harm of overestimation bias.
Istva ́n Szita and Andra ́s Lo ̋rincz. The Many Faces of Optimism: A Unifying Approach. In International Conference on Machine learning, 2008.

Alexander L. Strehl, Lihong Li, and Michael L. Littman. Reinforcement Learning in Finite MDPs: PAC Analysis. Journal of Machine Learning Research, 2009.

---

### Decision · Program_Chairs · 2021-01-07
**Final Decision**

**Decision:**

Reject

**Comment:**

The reviewers were in agreement that the paper is below the bar for acceptance, and the authors did not provide a response to reviewer concerns.